# Solid Nanocrystals of Rebamipide Promote Recovery from Indomethacin-Induced Gastrointestinal Bleeding

**DOI:** 10.3390/ijms20204990

**Published:** 2019-10-09

**Authors:** Noriaki Nagai, Ryusuke Sakamoto, Seiji Yamamoto, Saori Deguchi, Hiroko Otake, Tadatoshi Tanino

**Affiliations:** 1Faculty of Pharmacy, Kindai University, 3-4-1 Kowakae, Higashi-Osaka, Osaka 577-8502, Japan; 1511610012m@kindai.ac.jp (R.S.); 1611610133t@kindai.ac.jp (S.Y.); 1111610121m@kindai.ac.jp (S.D.); hotake@phar.kindai.ac.jp (H.O.); 2Faculty of Pharmaceutical Sciences, Tokushima Bunri University, 180 Yamashiro-Cho, Tokushima 770-8514, Japan; tanino@ph.bunri-u.ac.jp

**Keywords:** gastrointestinal injury, nanoparticle, rebamipide, indomethacin, rheumatoid arthritis

## Abstract

Indomethacin (IMC)-induced gastrointestinal (GI) injuries are more common in rheumatoid arthritis (RA) patients than in other IMC users, and the overexpression of nitric oxide (NO) via inducible NO synthase (iNOS) is related to the seriousness of IMC-induced GI injuries. However, sufficient strategies to prevent IMC-induced GI injuries have not yet been established. In this study, we designed dispersions of rebamipide (RBM) solid nanocrystals (particle size: 30–190 nm) by a bead mill method (RBM-NDs), and investigated whether the oral administration of RBM-NDs is useful to prevent IMC-induced GI injuries. The RBM nanocrystals were spherical and had a solubility 4.71-fold greater than dispersions of traditional RBM powder (RBM-TDs). In addition, the RBM-NDs were stable for 1 month after preparation. The RBM contents in the stomach, jejunum, and ileum of rats orally administered RBM-NDs were significantly higher than in rats administered RBM-TDs. Moreover, the oral administration of RBM-NDs decreased the NO levels via iNOS and area of the GI lesions in IMC-stimulated RA (adjuvant-induced arthritis rat) rats in comparison with the oral administration of RBM-TDs. Thus, we show that the oral administration of RBM-NDs provides a high drug supply to the GI mucosa, resulting in a therapeutic effect on IMC-induced GI injuries. Solid nanocrystalline RBM preparations may offer effective therapy for RA patients.

## 1. Introduction

Although nonsteroidal anti-inflammatory drugs (NSAIDs) are widely used to help relieve inflammation and musculoskeletal pain, adverse gastrointestinal (GI) events are increasingly prevalent following the widespread use of NSAIDs. It is reported that approximately 10–30% of chronic users of NSAIDs develop GI hemorrhages, erosion, peptic ulceration, and dyspepsia [1,2]. The mechanisms of adverse GI effects have been the focus of many studies. NSAIDs cause a deficiency of prostaglandin via the inhibition of a cyclooxygenase, resulting in a lowered defense system. In the stomach, the gastric mucosa is damaged by gastric acid and direct drug stimulation, resulting in gastric injury [3]. NSAID-induced small intestinal damage is also thought to be caused by prostaglandin deficiency. On the other hand, there are some pathophysiological differences between of the gastric and small intestinal damage done by NSAIDs, as the small intestinal damage is independent of gastric acid. Previous studies have reported that enterobacterial invasion causes an upregulation of nitric oxide (NO) via inducible nitric oxide synthase (iNOS), decreasing mucus secretion and intestinal hypermotility [4]. These multiple factors lead to the mucosal injuries in the small intestine. In addition, it is known that patients with rheumatoid arthritis (RA) are more sensitive in their NO production system via iNOS and interleukin (IL)-18 [5,6], and are reportedly more susceptible to NSAID-induced GI injuries when compared with other NSAID users [7,8].

Rebamipide (RBM), 2-(4-chlorobenzoylamino)-3-[2(1*H*)-quinolinon-4-yl] propionic acid, is a mucosal protective agent. The Biopharmaceutical Classification System (BCS) lists RBM as a class IV drug, and the pK and logP of RBM are 3.3 and 2.9, respectively. RBM enhances endogenous prostaglandins E2 and I2, which leads to anti-inflammatory action, mucin secretagogue activity, and antibacterial effects [9,10,11,12,13]. Moreover, RBM has an anti-inflammatory action, scavenges free radicals, suppresses increases in permeability, and improves blood flow [14,15,16,17]. On the basis of these features, RBM is widely used as therapy for GI disorders [18], and current evidence shows that RBM is effective and safe for defending against NSAID-induced GI injuries [19]. Therefore, RBM oral formulations are highly anticipated for the treatment NSAID-induced GI injuries.

There have been many studies on ways to enhance the usefulness and bioavailability of drugs, including a focus on the advantages of nanoparticles. Formulations using a core-shell-type polymeric micelle based on the self-assembly of methoxy-poly(ethylene glycol)-β-poly[4-(2,2,6,6-tetramethylpiperidine-1-oxyl) oxymethyl-styrene] [20] and a biodegradable polymer PLGA [poly(dl-lactide-co-glycolide)] [21,22,23,24] have been reported, and these formulations show high bioavailability, as well as therapeutic effects that are also better than those of traditional formulations. We have designed drug nanocrystals of indomethacin (IMC), ketoprofen [25], cilostazol [26,27], and tranilast [28], and have reported that these nanocrystals show increased drug solubility and enhanced bioavailability in the intestine [29]. In addition, we reported that nanocrystals in the range of 50–200 nm in size are taken up into the intestinal epithelium by endocytosis, such as clathrin-dependent endocytosis and caveolae-dependent endocytosis, and that the drug contents in cells reach higher levels than those that are reached with the larger 1 μm drug powder (traditional powder) [30]. It is possible that the application of this strategy to RBM may enhance its therapeutic effects on NSAID-induced GI injuries. In this study, we designed dispersions consisting of RBM solid nanocrystals (RBM-NDs), and demonstrated their usefulness against NSAID-induced GI injuries.

## 2. Results

### 2.1. Development of RBM-NDs

A wet mill is useful for the preparation of nanocrystals. Therefore, we prepared RBM nanocrystals by wet milling using zirconia balls and some additives, and evaluated the characteristics of the RBM in RBM-NDs. The RBM particle sizes in the dispersion consisting of conventional RBM (traditional RBM, RBM-TDs) ranged from 0.28 to 15 μm (Figure 1A), and this particle size was decreased by the bead mill treatment. The RBM particle sizes in the RBM dispersions treated with the bead mill (RBM-NDs) were in the range of 30–190 nm, and the distribution of RBM was less (Figure 1B–D), and the particles appeared spherical (Figure 2B). Figure 2A shows the XRD data. Solid RBM in the RBM-NDs remained in the crystal form (Figure 2A) with a crystal structure the same as in the RBM-TDs (data not shown). The zeta potentials were similar between RBM-TDs and RBM-NDs with levels of approximately -9 mV (Figure 2C). On the other hand, RBM solubility was enhanced by bead mill treatment. The RBM solubilities for RBM-TDs and RBM-NDs were 82.5 and 388.3 μM, respectively (Figure 2D). Figure 3 shows the changes in characterization (particle size and concentration) in RBM-NDs 1 month after preparation. Although the particle size distribution in the RBM-NDs had spread, the RBM particles remained in the nano size order (30–220 nm) 1 month after preparation (Figure 3A,C). Moreover, the number of RBM nanoparticles in RBM-NDs did not change (Figure 3B), and no difference in the concentration was observed 1 month after preparation (Figure 3D).

### 2.2. Drug Retention in GI Mucosa after the Oral Administration of RBM-NDs

Our previous study showed that drug retention in the GI mucosa is increased by nanonization [31], and it is possible that this enhanced retention leads to locally more effective therapy in the GI tract. Therefore, the RBM contents in the GI tract of rats orally administered RBM dispersions were measured in this study (Figure 4). The RBM contents in the gastric mucosa of rats orally administered RBM-TDs reached a maximum 3 h after administration, and 6 h after administration only a small amount of RBM remained. In rats orally administered RBM-NDs, the RBM contents in the gastric mucosa also reached a peak 3 h after administration, but RBM contents remained detectable for up to 24 h after administration (Figure 4A). Figure 4C shows the RBM contents in the jejunal mucosa of rats orally administered RBM-TDs and RBM-NDs. In both cases, RBM remained detectable in the jejunal mucosa for 24 h, but the RBM contents in the rats that received orally administered RBM-NDs were significantly higher than in rats administered RBM-TDs. On the other hand, the RBM contents in the ileal mucosa of rats orally administered with RBM-TDs and RBM-NDs were similar during the period of 0–6 h after administration (Figure 4E); however, 24 h after administration, the RBM contents in rats orally administered RBM-NDs was 2.5-fold higher than in rats administered RBM-TDs (Figure 4E). Figure 4B,D,F shows the areas under the drug concentration-time curves (*AUC*_0–24h_) in the gastric, jejunal, and ileal mucosa of rats orally administered RBM-TDs and RBM-NDs. The *AUC*_0–24h_ for rats administered RBM-NDs were significantly higher than those for rats administered RBM-TDs.

### 2.3. Therapeutic Effect of RBM-NDs on IMC-Induced GI Bleeding

It is known that the oral administration of IMC causes GI bleeding, and serious GI bleeding is observed in adjuvant-induced arthritis (AA) rats in comparison with normal rats [7,8]. Therefore, we prepared AA rats and investigated whether the oral administration of RBM can reduce their GI bleeding. Figure 5 shows an image of the stomach, jejunum, and ileum of rats orally administered excessive IMC (40 mg/kg). GI bleeding in the stomach, jejunum, and ileum was observed in both IMC-stimulated normal and AA rats, but the ulcerogenic lesions were more serious in the IMC-stimulated AA rats. Figure 6 shows the therapeutic effect of RBM dispersions on IMC-induced GI bleeding in AA rats. The oral administration of RBM-TDs promoted the recovery from GI bleeding, and levels of lesions in the gastric, jejunal, and ileal mucosas were found to be 61.4%, 63.3%, and 84.6% in the RBM non-treated group, respectively. The oral administration of RBM-NDs strongly decreased the GI lesion areas in AA rats, having a therapeutic effect significantly higher than that of RBM-TDs. GI lesions in the gastric, jejunal, and ileal mucosas of rats orally administered RBM-NDs were 35.7%, 23.9%, and 19.7% of those in the case of RBM-TD administration, respectively. Figure 7 shows the iNOS mRNA and NO levels in IMC-stimulated AA rats. Both levels were higher in the RBM non-treated groups when compared with the RBM treated groups. In addition, the oral administration of RBM-NDs significantly suppressed the expression of iNOS and NO in the gastric, jejunal, and ileal mucosas when compared with groups treated with vehicle or RBM-TDs.

## 3. Discussion

It is known that the IMC, one of the NSAIDs, causes serious GI side effects including dyspepsia, and that IMC-induced GI injuries are more developed RA patients than in other IMC users [1,2]. However, a sufficient strategy to prevent IMC-induced GI injuries has not been established thus far. Therefore, there is an urgent need to develop novel treatment approaches for IMC-induced GI damage in the therapy of RA. Recently, it was reported that RBM is safe and effective in treating NSAID-induced GI injuries, and that oral RBM formulations are expected to have a therapeutic effect on NSAID-induced GI injuries. In this study, we prepared RBM nanocrystals (30–190 nm), and found that the oral administration of RBM-NDs resulted in high drug retention in the mucosa of the stomach, jejunum, and ileum. In addition, we show that the oral administration of RBM-NDs promoted the recovery of the GI lesion area in AA rats stimulated with excessive IMC.

First, we attempted to obtain solid nanocrystals of RBM. Various methods have been reported to prepare nanocrystals, of which bottom-up synthesis and a top-down approach are two major approaches. Bottom-up synthesis is based on emulsion systems and self-assembly. A top-down approach means reducing the particle size from large to smaller. Our previous studies have shown that the combination of certain additives and the bead mill method, a top-down approach, can prepare high quality dispersions based on drug solid nanocrystals without harmful solvents [30]. Therefore, an approach involving the bead mill method in the presence of with two main additives (methylcellulose (MC) to enhance mill efficiency and 2-hydroxypropyl-β-cyclodextrin (HPβCD) to prevent the aggregation) was selected according to our previous reports [32]. After treatment with the combination of additives and bead mill, the ratio of dissolved RBM was 0.003% in RBM-NDs, and 98.6% of the RBM was present as solid nanocrystals (Figure 2D) with a particle size in the range of 30 to 190 nm (Figure 1B,C). It is known that both HPβCD and nanonization relate to the increase in drug solubility. In this study, we also measured the effect of HPβCD on the solubility, and the solubility in RBM-TDs and RBM-NDs without HPβCD were 61.8 ± 5.1 μM and 216.5 ± 11.9 μM, respectively (*n* = 6). From these results, it was suggested that the combination of milling and HPβCD was related to the enhancement of REB solubility in the RBM-NDs. Moreover, the RBM particles in RBM-NDs remained in the nano size (30–220 nm) 1 month after production (Figure 3A). These results show that the RBM nanocrystals prepared by the bead mill method are stable.

Next, we investigated the therapeutic effect of RBM-NDs on NSAID-induced GI injuries. In the therapy of GI injuries, RBM retention in the GI mucosa is related to drug efficacy after oral administration. RBM contents in the gastric, jejunal, and ileal mucosa of rats that were orally administered RBM-NDs were significantly higher when compared with rats administered RBM-TDs (Figure 4). In addition, our previous study using cilostazol also showed that nanonization increased drug adhesiveness in the mucosa and enhanced retention in the GI mucosa after oral administration [31]. The results shown in Figure 4 support the previous studies.

In order to study the therapeutic effects of RBM on NSAID-induced GI injuries in RA patients, it is important to choose an appropriate animal model. RA is a chronic autoimmune disease associated with increases in tumor necrosis factor α, IL-6, IL-17, IL-18, rheumatoid factor, and other pro-inflammatory cytokines [5,6,33,34,35]. The onset of RA causes tenderness, joint swelling, synovial hyperplasia, and chronic inflammation, and the one therapeutic approach for RA used to alleviate pain is through NSAIDs. The AA rat is an animal model for RA. Paw volume reflects inflammatory pain in AA rats [36,37], and systemic inflammation is observed 2 weeks after the injection of adjuvant [6,38]. Moreover, the NSAID-induced GI injuries in AA rats are aggravated in comparison with normal rats (non-AA rats) [6,38]. Thus, the AA rat is a useful model in which to test therapeutic effects against NSAID-induced GI injuries in RA patients. In this study, lesions in the gastric, jejunal, and ileal mucosa were caused by excessive IMC stimulation, and the levels of GI lesions were significantly higher in AA rats when compared with normal rats (Figure 5 and Figure 6). In addition, the oral administration of RBM-NDs strongly decreased the lesion areas in the gastric, jejunal, and ileal mucosa in IMC-stimulated AA rats, and this therapeutic effect was significantly higher than that of RBM-TDs (Figure 6). Several reports have shown that RBM provides mucosal protection in IMC-induced gastric injuries through radical scavenging action, prostaglandin biosynthesis, and/or the enhancement of mucus secretion [39,40,41]. Moreover, the RBM alters the composition of the gut microbiota in mice administered IMC, resulting in an attenuation of IMC-induced small intestinal injuries [42]. It has also been reported that RBM inhibits the enhanced NO production via iNOS activity induced by excessive IMC, thereby reducing IMC-induced GI injuries in rats [43]. Therefore, we further demonstrated that the oral administration of RBM-NDs attenuates the overproduction of NO by iNOS in IMC-induced GI injuries. Both the iNOS mRNA and NO levels in IMC-stimulated AA rats were increased in comparison with normal rats, and the enhanced iNOS mRNA and NO levels were attenuated by treatment with RBM-TDs (Figure 7). Moreover, RBM-NDs significantly decreased the overexpression of the iNOS mRNA and NO production to a greater extent than RBM-TDs (Figure 7). Taken together, we hypothesise that the excessive production of iNOS and NO by IMC causes injuries in the gastric, jejunal, and ileal mucosa of AA rats, and the oral administration of RBM-NDs promotes the repair of IMC-induced GI injuries, and attenuates NO production in comparison with RBM-TDs, as the retention time and amounts of RBM in the GI mucosa of rats orally administered RBM-NDs are higher than for rats administered RBM-TDs.

Further studies are needed to clarify the precise mechanisms of the therapeutic effects of RBM solid nanocrystals in IMC-induced GI injuries. We plan to use immunohistochemistry to observe the GI condition in rats treated with RBM. Moreover, it is important to elucidate the uptake mechanism of RBM solid nanocrystals into tissue. Our previous studies showed that particles up to approximately 100 nm in size are taken up into cells as solids, and that energy-dependent endocytosis is related to this uptake [30,44]. Furthermore, drug uptake by energy-dependent endocytosis increases bioavailability, resulting in an enhancement in plasma drug levels [44]. Therefore, we are investigating the contribution of energy-dependent endocytosis and enhanced plasma RBM in the therapeutic effect of IMC-induced GI injuries.

## 4. Materials and Methods 

### 4.1. Drug and Chemicals

*N*,*N*-dimethylformamide, conventional RBM, and IMC powder were provided by Wako Pure Chemical Industries Ltd., (Osaka, Japan). Membrane filters with 25 nm pore size (nitrocellulose membranes) were purchased from Merck Millipore Ltd. (Billerica, MA, USA). MC and HPβCD were obtained from Shin-Etsu Chemical Co. Ltd. (Tokyo, Japan) and Nihon Shokuhin Kako Co. Ltd. (Tokyo, Japan), respectively. Heat-killed *Mycobacterium butyricum* was purchased from Difco (Detroit, MI, USA). The Protein Assay Kit was obtained from BIO-RAD (Hercules, CA, USA). Other reagents were of the highest purity commercially available.

### 4.2. Experimental Animals

The severity of arthritis in dark agouti (DA) rats is higher than in other rats, and the incidence of arthritis development is 100% [38]. Therefore, DA rats were used in this study. Male DA rats aged 6 weeks were obtained from Shimizu Laboratory Supplies Co., Ltd. (Kyoto, Japan), and were kept under standard laboratory conditions on a standard diet. The experimental study was approved by the School of Pharmacy Committee for the Care and Use of Laboratory Animals at Kindai University (project identification code KAPS-25-004, 1 April 2013). In addition, rats were used in accordance with the guiding principles approved by the Japanese Pharmacological Society and the guidelines for animal experimentation of the International Association for the Study.

### 4.3. Design of RBM Solid Nanocrystals

Solid RBM nanocrystals were prepared according to our previous study [29]. A total of 1% conventional RBM powder and 0.5% MC were dispersed in 5% HPβCD solution (traditional RBM dispersions, RBM-TDs). Then, 100 μm zirconia balls were added to the RBM-TDs, and the mixtures were subjected to repetitive milling at 5500 rpm at 4 °C (30 s/time, 15 times). The dispersions with the milled-RBM were used as dispersions consisting of RBM solid nanocrystals (RBM-NDs). The pH of the RBM-TDs and RBM-NDs was adjusted to 7.

### 4.4. Powder X-ray Diffraction (XRD)

RBM was dispersed in water and milled in the presence of zirconia balls. Afterwards, the milled-RBM was lyophilized. Mini Flex II was used to analyze the crystal form of the lyophilized RBM (Rigaku Co., Tokyo, Japan) under conditions as follows: scanning rate, 10°/min; diffraction angles, 5° to 90°; X-rays, 30 kV and 15 mA.

### 4.5. Characterization in RBM Solid Nanocrystals

RBM-TDs and RBM-NDs were stirred for 24 h at 25 °C and centrifuged at 100,000× *g*, and the supernatants were filtered through 25 nm pore size membrane filters. RBM content in the filtrates was measured to evaluate drug solubility by an HPLC method (stability test). Fifty microliters of RBM was injected with an auto sampler CTO-20A, and RBM in the samples was detected at 254 nm by a Shimadzu LC-20AT system (Shimadzu Corp., Kyoto, Japan). An Inertsil ODS-3 (3 μm) column (GL Science Co., Inc., Tokyo, Japan) was used, and the column temperature was maintained at 35 °C using a column oven CTO-20A. The mobile phase was acetonitrile/phosphate buffer (17/83, *v*/*v*) at a flow rate of 250 μL/min. The zeta potential of the RBM particles was determined by a Zeta Potential Meter Model 502 (Nihon Rufuto Co., Ltd., Tokyo, Japan), and atomic force microscopic (AFM) images of RBM nanocrystals were created by a combination of a phase and height image using image analysis software connected to a scanning probe microscope (SPM)-9700 (Shimadzu Corp., Kyoto, Japan). A laser diffraction particle size analyzer SALD-7100 (Shimadzu Corp., Kyoto, Japan) was used to measure the particle size distribution and frequency of the RBM-TDs and RBM-NDs preparations, and the complex refractive index used for measuring was set at 1.60–0.010i. In addition, the size distribution and number of RBM particles in RBM-NDs were also determined with a NANOSIGHT LM10 (QuantumDesign Japan, Tokyo, Japan) under the following measuring conditions: viscosity, 0.904–0.906 mPa⋅s; wavelength, 405 nm; measurement time, 60 s.

### 4.6. Model of IMC-Induced GI Injuries

Rats were injected with 50 μL Bayol F oil containing 10 mg/mL of heat-killed *Mycobacterium butyricum* into both the right hind foot and tail, and were kept under standard laboratory conditions for 14 days. The *Mycobacterium butyricum*-injected rats showed the onset of systemic inflammation, and were used as an arthritis model (AA rat). The inflammation was evaluated by measuring the paw volume by plethysmometry (right hind foot 3.0 ± 0.7 mL, left 2.5 ± 0.6 mL, *n* = 20). GI bleeding was induced by the oral administration of excessive IMC (40 mg/kg). To induce gastric mucosal lesions, the IMC was administered to 12 h fasted rats. On the other hand, to study the IMC-induced injuries in the jejunal and ileal mucosa, the IMC was administered to non-fasted rats. In this study, 40% of the upper part of the small intestine was excised as the jejunum (27.4 ± 2.9 cm, *n* = 20) and 60% of the lower part of the small intestine was used as the ileum (39.3 ± 5.7 cm, *n* = 20).

### 4.7. Therapeutic Effect of RBM in Model Rats with IMC-Induced GI Injuries

RBM dispersions were administered orally at a dose of 2 mg/kg. In the experiment to test the therapeutic effects of RBM on IMC-induced gastric injuries, the RBM was administered 6 h after excessive IMC administration, and the rats were killed under deep isoflurane anesthesia 18 h later. In the experiment to test the effects of RBM on IMC-induced jejunal and ileal injuries, the RBM was administered 24 h after excessive IMC administration, and the rats were killed under deep isoflurane anesthesia 24 h later. The stomach, jejunal, and ileal mucosas were excised and fixed in 10% formalin solution, and the lesion areas were observed in digital photographs and measured with ImageJ. In this study, the areas of injury were expressed as ratios of the lesion areas in the total mucosal area (%).

### 4.8. RBM Contents in the Gastric, Jejunal, and Ileal Mucosa after Oral Administration of RBM

RBM was orally administered to 8 h fasted rats (2 mg/kg). Then, 3, 6, or 24 h later the rats were killed under deep isoflurane anesthesia, and the gastric, jejunal, and ileal mucosas were excised. The mucosal samples were homogenized in *N*,*N*-dimethylformamide on ice to extract the RBM, and the supernatants were collected by centrifugation at 20,400× *g* for 15 min at 4 °C. The RBM contents were evaluated by the HPLC method described above. The protein levels in the samples were analyzed by the Protein Assay Kit, and the RBM contents in the gastric, jejunal, and ileal mucosas were expressed as microgram per milligram protein. The area under the drug concentration time curve (*AUC*_0–24h_) was analyzed according to the trapezoidal rule for 0–24 h.

### 4.9. Real Time Polymerase Chain Reaction (PCR)

The rats were killed under deep isoflurane anesthesia, and the gastric, jejunal, and ileal mucosas were excised. A LightCycler DX 400 was used for real time PCR (Roche Diagnostics Applied Science, Mannheim, Germany) [5,6]. Briefly, an acid guanidium thiocyanate-phenol-chloroform extraction method was applied to extract total RNA from the gastric, jejunal, and ileal mucosal tissues, and the reverse transcriptase (RT) and PCR reactions were performed using an RNA PCR Kit (TaKaRa Bio Inc., Shiga, Japan) and LightCycler FastStart DNA Master SYBR Green I (Roche Diagnostics Applied Science, Mannheim, Germany) according to the manufacturers’ instructions. Glyceraldehyde-3-phosophate dehydrogenase (GAPDH) or iNOS primers were used as follows: 5-ACGGCACAGTCAAGGCTGAGA-3 and 5-CGCTCCTGGAAGATGGTGAT-3 for GAPDH (NM_017008), and 5-GGAGAGATTTTTCACGACACCC-3 and 5-CCATGCATAATTTGGACTTGCA-3 for iNOS (NM_012611). The PCR conditions for denaturing, annealing, and extension were 95 °C for 10 s, 60 °C for 10 s, and 72 °C for 5 s, respectively. The iNOS mRNA levels were expressed as the iNOS/GAPDH.

### 4.10. Measurement of NO Levels

The rats were killed under deep isoflurane anesthesia, and the gastric, jejunal, and ileal mucosas were excised and homogenized in saline on ice. The homogenates were then centrifuged at 20,400× *g* for 10 min at 4 °C, and the supernatants were collected. The NO levels were measured using ENO-20 (Eicom, Kyoto, Japan) according to our previous study [5,6]. Briefly, ions of nitrous acid (NO_2_^−^) and nitrate (NO_3_^−^) were separated on a NO-PAK column (4.6 × 50 mm, Eicom, Kyoto, Japan), and NO_2_^−^ was measured with Griess reagent and a NOD-10 (Eicom, Kyoto, Japan) at 540 nm. The levels of NO_2_^−^ metabolite produced from NO were expressed as NO levels.

### 4.11. Statistical Analysis

Statistical analyses were done by Student’s *t*-test and Dunnett’s multiple comparison, with *p* < 0.05 considered statistically significant. All data points were expressed as the mean ± standard error of the mean (S.E.).

## 5. Conclusions

We designed solid nanocrystals of RBM (30–190 nm), and found that the oral administration of RBM nanocrystals resulted in higher retention and contents in the GI mucosa when compared with the oral administration of RBM-TDs. In addition, the RBM nanocrystals suppressed the expression of iNOS and NO, and provided an enhanced therapeutic effect in the GI of IMC-stimulated AA rats in comparison with RBM-TDs. These results provide significant information concerning therapy against NSAID-induced GI injuries in RA patients.

## Figures and Tables

**Figure 1 ijms-20-04990-f001:**
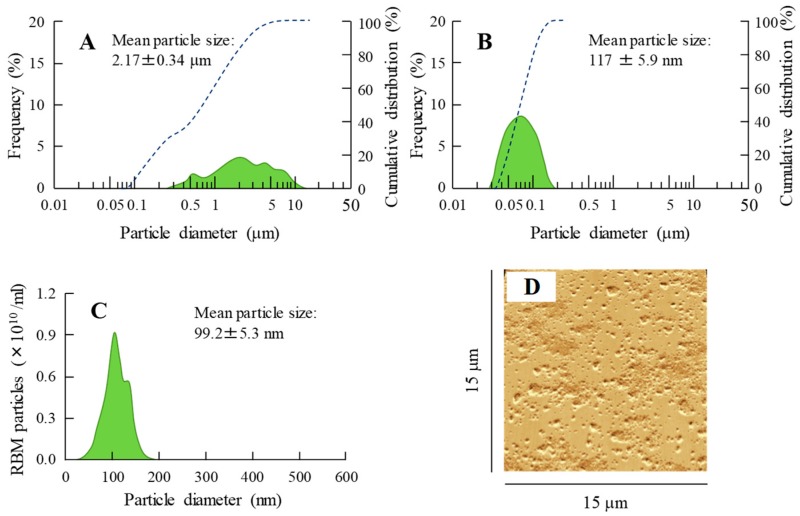
Changes in particle size by nanonization of rebamipide (RBM) using the bead mill treatment. Particle size frequencies (solid line, green) and cumulative distribution (dashed line, blue) of RBM in traditional RBM powder (RBM-TDs) (**A**) and RBM solid nanocrystals (RBM-NDs) (**B**) as determined by a laser diffraction particle size analyzer. (**C**) Particle size frequencies of RBM in RBM-NDs by the dynamic light scattering method. (**D**) Atomic force microscopy (AFM) image of RBM in RBM-NDs (wide-field). The scale of depth and width are 15 μm. These experiments were repeated six times. The RBM particle size in RBM-NDs was in the range of 30–190 nm.

**Figure 2 ijms-20-04990-f002:**
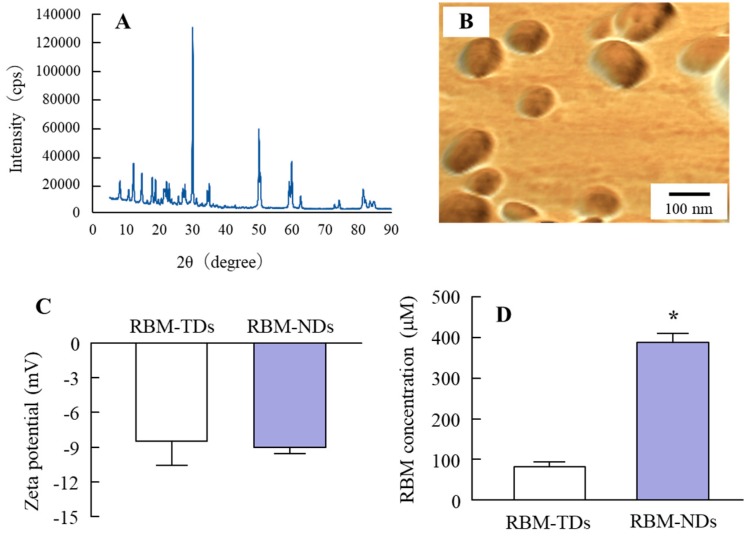
Morphology and particle characteristics of RBM in RBM-NDs. (**A**) XRD pattern of RBM treated with a bead mill. (**B**) AFM image of RBM in RBM-NDs. (**C**) Zeta potential of RBM in RBM-NDs. (**D**) Solubility of RBM in RBM-NDs. *n* = 6, * *p* > 0.05 vs. RBM-TDs. The RBM remained in the crystal form after treatment with the bead mill, and the form was spherical. The solubility of RBM was increased by bead mill treatment, and the content of dissolved RBM in RBM-NDs was 4.7-fold higher than in RBM-TDs. On the other hand, there was no difference in the zeta potential between RBM-TDs and RBM-NDs.

**Figure 3 ijms-20-04990-f003:**
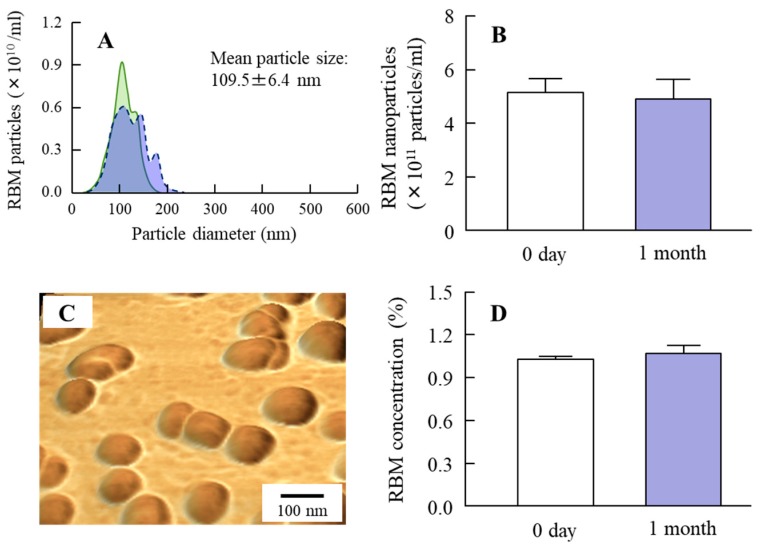
Changes in particle character and concentration of RBM-NDs 1 month after the preparation. (**A**) Particle size frequencies of RBM in RBM-NDs 0 days (solid line, green) and 1 month (dashed line, blue) after the preparation. (**B**) Changes in particle number of RBM nanoparticles in RBM-NDs. (**C**) AFM image of RBM in RBM-NDs. (**D**) RBM concentration in RBM-NDs. *n* = 6. The particle in RBM-NDs remained nano size (30–220 nm) 1 month after the production, and no difference in particle number, form, or concentration were observed between 0 days and 1 month after preparation.

**Figure 4 ijms-20-04990-f004:**
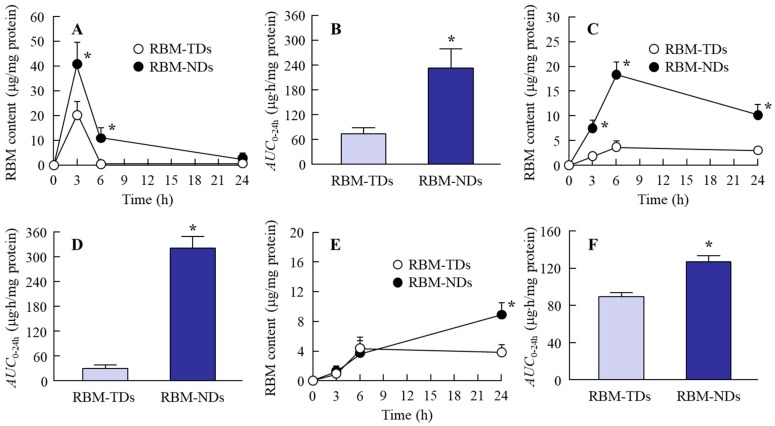
Changes in RBM contents in the stomach, jejunum, and ileum of rats orally administered RBM-TDs and RBM-NDs. RBM contents (**A**) and areas under the drug concentration-time curves (AUC)_0–24h_ (**B**) in the stomach. RBM contents (**C**) and *AUC*_0–24h_ (**D**) in the jejunum. RBM contents (**E**) and *AUC*_0–24h_ (**F**) in the ileum. RBM-TDs: RBM-TDs-administered rats; RBM-NDs: RBM-NDs-administered rats. *n* = 6–9, * *p* > 0.05 vs. RBM-TDs. In the stomach, jejunum, and ileum, the RBM contents in the rats that received orally administered RBM-NDs were significantly higher when compared with rats administered RBM-TDs.

**Figure 5 ijms-20-04990-f005:**
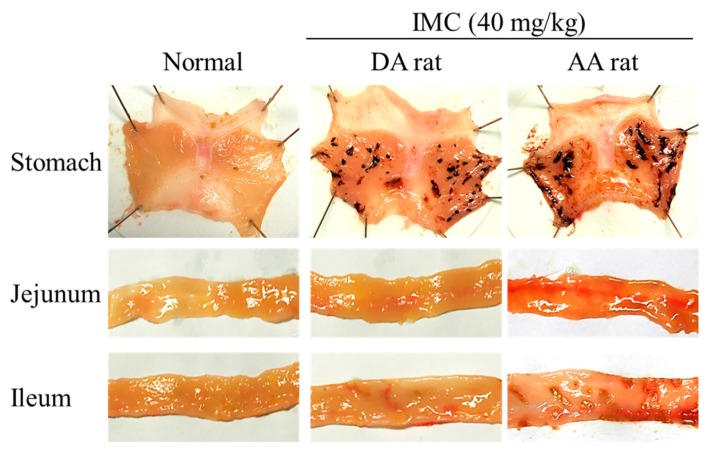
Representative images of indomethacin (IMC)-induced gastrointestinal (GI) bleeding in normal and adjuvant-induced arthritis (AA) rats. The images were obtained 24 h (stomach) or 48 h (jejunum and ileum) after the oral administration of excessive IMC (40 mg/kg). Normal: normal dark agouti (DA) rat (non-treatment); IMC + DA rat: IMC-stimulated DA rat; IMC + AA rat: IMC-stimulated AA rat. In the stomach, jejunum, and ileum, GI bleeding was caused by excessive IMC stimulation, and the levels of GI lesions in the AA rats were significantly higher than in DA rats (non-AA rats).

**Figure 6 ijms-20-04990-f006:**
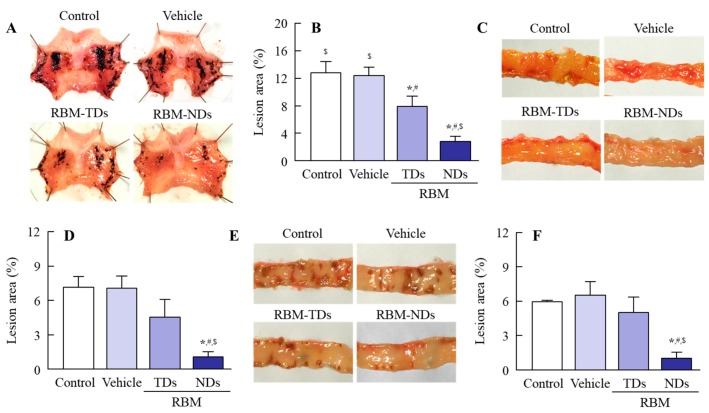
Therapeutic effect of RBM-NDs on IMC-induced lesions in the gastric, jejunal, and ileal mucosa of AA rats. (**A**,**C**,**E**) Representative images of stomach (**A**), jejunum (**C**), and ileum (**E**) of rats treated with vehicle, RBM-TDs, and RBM-NDs. (**B**,**D**,**F**) Changes in gastric (**A**), jejunal (**D**), and ileal (**F**) lesion areas in rats treated with vehicle, RBM-TDs, and RBM-NDs. Control: IMC-stimulated AA rat; vehicle: IMC-stimulated AA rat treated with vehicle; RBM-TDs: IMC-stimulated AA rat treated with RBM-TDs; RBM-N: IMC-stimulated AA rat treated with RBM-NDs. *n* = 7–11, * *p* > 0.05 vs. control. ^#^
*p* > 0.05 vs. vehicle, ^$^
*p* > 0.05 vs. RBM-TDs. The oral administration of RBM-NDs decreased the GI lesion area in IMC-stimulated AA rats, and the therapeutic effect was higher than that of RBM-TDs.

**Figure 7 ijms-20-04990-f007:**
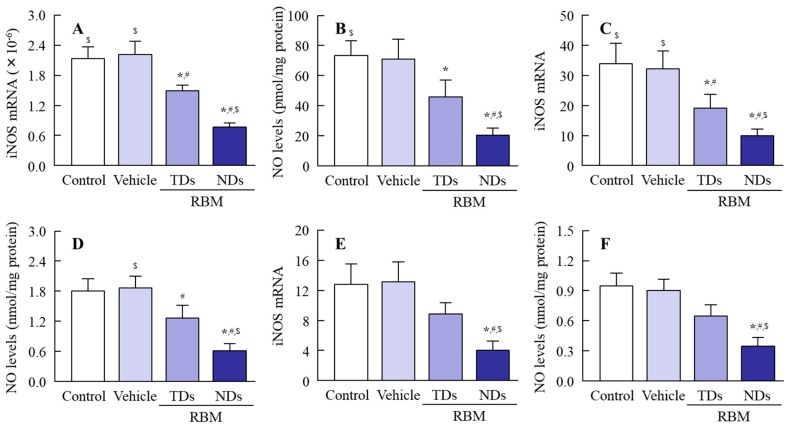
Preventive effect of RBM-NDs on increased levels of inducible nitric oxide synthase (iNOS) mRNA and nitric oxide (NO) in the gastric, jejunal, and ileal mucosa of IMC-stimulated AA rats. (**A**,**C**,**E**) iNOS mRNA in the gastric (**A**), jejunal (**C**), and ileal (**E**) mucosa in AA rats treated with vehicle, RBM-TDs, and RBM-NDs. (**B**,**D**,**F**) NO levels in the gastric (**B**), jejunal (**D**), and ileal (**F**) mucosa in AA rats treated with vehicle, RBM-TDs, and RBM-NDs. Control: IMC-stimulated AA rats; vehicle: IMC-stimulated AA rats treated with vehicle; RBM-TDs: IMC-stimulated AA rats treated with RBM-TDs; RBM-N: IMC-stimulated AA rats treated with RBM-NDs. *n* = 4–9, * *p* > 0.05 vs. control, ^#^
*p* > 0.05 vs. vehicle, ^$^
*p* > 0.05 vs. RBM-TDs. RBM attenuated the overexpression of iNOS mRNA and NO, and the preventive effects were greater for RBM-NDs than RBM-TDs.

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
