# Peer review of "Solid Nanocrystals of Rebamipide Promote Recovery from Indomethacin-Induced Gastrointestinal Bleeding"

_ijms, 2019, doi:10.3390/ijms20204990_

Round 1

Reviewer 1 Report

The manuscript describes the preventive effect of rebamipide nanoparticles on indomethacin-induced gastrointestinal injuries. The article is consistent and well written and can be accepted for publication in Int. J. Mol. Sci. after the minor revision.

Minor point

1) Concerning the drug, rebamipide, please add information in terms of pK, logP, permeability, BCS system, etc

2) Why the N,N-dimethylformamide were used to extract the rebamipide from tissue.

3) Based on the words in results, the Figure 3C is obtained through the analysis of AFM imaging. Could the authors provide the detailed method for that? Is it automated done by the AFM or calculated based on some AFM pictures?

4) line 267: please correct "DS rats" to DA rats.

Line 306: It should be "mycobacterium" from “Mycobacterium”.

Author Response

We carefully revised our manuscript according to the suggestions of the reviewer 1, and details are as follows.

Q1. Concerning the drug, rebamipide, please add information in terms of pK, logP, permeability, BCS system, etc

A1. The reviewer’s comments are very important. The pK and logP of rebamipide are 3.3, 2.9, respectively. Biopharmaceutical Classification System (BCS) lists rebamipide as Class IV drug. In order to respond to the reviewer’s comment, we added the information in the Introduction (line 49-50).

Q2. Why the N,N-dimethylformamide were used to extract the rebamipide from tissue

A2. Although, the solubility of rebamipide is very low in the methanol, ethanol and water, the N,N-dimethylformamide can dissolve the rebamipide well. Therefore, we selected the N,N-dimethylformamide to extract from the tissue. Thank you for pointing out this (line 338).

Q3. Based on the words in results, the Figure 3C is obtained through the analysis of AFM imaging. Could the authors provide the detailed method for that? Is it automated done by the AFM or calculated based on some AFM pictures?

A3. Thank you very much for pointing this out. AFM images of rebamipide nanocrystals was created by combination of a phase and height image using image analysis software connected to the scanning probe microscope (SPM)-9700 (Shimadzu Corp., Kyoto, Japan). The experiment was done 6 time, and the reproducibility of that is sufficient. In order to respond to the reviewer’s comment, we added the information in the Materials and methods (line 96, 304, 305).

Q4. line 267: please correct "DS rats" to DA rats.

Line 306: It should be "mycobacterium" from “Mycobacterium”.

A4. The reviewer’s comment is correct. In order to respond to the reviewer’s comment, we corrected to “DA rat”. In addition, we revised to “mycobacterium” (line 274, 314, 315).

Reviewer 2 Report

The authors performed a good study, however, there are still some issues needed to be solved.

AFM is not a good method to characterize nanoparticles with bias. The author should use SEM or TEM. SEM would be better to give us a whole picture of the distribution. I do not think the RBM solubility is only improved by wet milling. In fact HPβCD also helps a lot. The author should change the expression. The stability testing was performed in what condition? The author always mentioned that the nanonization improved everything. However, HPβCD is often used as a additive in drug formulation. So whether the HPβCD took the effect? Whether HPβCD did not affect anything? The language should be polished.

Author Response

We carefully revised our manuscript according to the suggestions of the reviewer 2, and details are as follows.

Q1. AFM is not a good method to characterize nanoparticles with bias. The author should use SEM or TEM. SEM would be better to give us a whole picture of the distribution.

A1. Thank you very much for pointing this out. It is important to show the whole picture of the distribution, and the data in SEM is better to get a whole picture of the distribution. On the other hand, the AFM is also able to provide the whole picture of the distribution by regulation of the measurement area (scale). Therefore, we measured the whole picture of REB-NPs by the AFM with wide-field. In order to respond to the reviewer’s comment, we added the image data in the Fig. 1D (79, 95, 96, Figure 1D).

Q2. I do not think the RBM solubility is only improved by wet milling. In fact HPβCD also helps a lot. The author should change the expression. The stability testing was performed in what condition? The author always mentioned that the nanonization improved everything. However, HPβCD is often used as a additive in drug formulation. So whether the HPβCD took the effect? Whether HPβCD did not affect anything? The language should be polished.

A2. The reviewer’s comments are very important. RBM-TDs and RBM-NDs were stirred for 24 h at 25ºC, and centrifuged at 100,000 g and the supernatants were filtered through 25 nm pore size membrane filters. RBM contents in the filtrates was measured to evaluate drug solubility by an HPLC method. The solubility in RBM-TDs and RBM-NDs without HPβCD were 61.8 ± 5.1 mM, 216.5 ± 11.9 mM, respectively (n=6). The result showed that the combination of milling and HPβCD related the enhancement of REB solubility. In order to respond to the reviewer’s comment, we added the data and information, and revised these sentence (103, 209-213, 295-297).

Round 2

Reviewer 2 Report

Good enough for publication.